# Sub-type specific connectivity between CA3 pyramidal neurons may underlie their sequential activation during sharp waves

**Rosanna P Sammons[1†], Stefano Masserini[2,3,4†], Laura Moreno Velasquez[1], Verjinia D Metodieva[1,4], Gaspar Cano[2], Andrea Sannio[1,4], Marta Orlando[1], Nikolaus Maier[1], Richard Kempter[2,3,4], Dietmar Schmitz[1,3,4,5,6,7]★**

[1]Charité-Universitätsmedizin Berlin, corporate member of Freie Universität Berlin and Humboldt-Universität zu Berlin, Neuroscience Research Center, Berlin, Germany; [2]Institute for Theoretical Biology, Department of Biology, Humboldt-Universität zu Berlin, Berlin, Germany; [3]Bernstein Center for Computational Neuroscience, Berlin, Germany; [4]Charité-Universitätsmedizin Berlin, corporate member of Freie Universität Berlin and Humboldt-Universität zu Berlin, Einstein Center for Neurosciences, Berlin, Germany; [5]German Center for Neurodegenerative Diseases (DZNE), Berlin, Germany; [6]Max-Delbrück Center for Molecular Medicine in the Helmholtz Association, Berlin, Germany; [7]Charité – Universitätsmedizin Berlin, corporate member of Freie Universität Berlin and Humboldt-Universität zu Berlin, NeuroCure Cluster of Excellence, Berlin, Germany

*For correspondence: dschmitz-office@charite.de

†These authors contributed equally to this work

## eLife Assessment

This study represents **valuable** findings on the asymmetric connectivity pattern of two different types of CA3 pyramidal cell types showing that while athorny cells receive strong inputs from all other cell types, thorny cells receive weaker inputs from athorny neurons. Computational modeling is used to evaluate the impact of this connectivity scheme on the sequential activation of different cell types during sharp wave ripples. The evidence combining experimental and computational modelling approaches **convincingly** supports the authors' claims regarding the network mechanisms underlying the temporal sequences of neuronal activity during sharp-waves.

**Abstract** The CA3 region of the hippocampus is the major site of sharp wave initiation, and a brain region crucially involved in learning and memory. Highly recurrent connectivity within its excitatory network is thought to underlie processes involved in memory formation. Recent work has indicated that distinct subpopulations of pyramidal neurons within this region may contribute differently to network activity, including sharp waves, in CA3. Exactly how these contributions may arise is not yet known. Here, we disentangle the local connectivity between two distinct CA3 cell types in mice: thorny and athorny pyramidal cells. We find an asymmetry in the connectivity between these two populations, with athorny cells receiving strong input from both athorny and thorny cells. Conversely, the thorny cell population receives very little input from the athorny population. Computational modeling suggests that this connectivity scheme may determine the sequential activation of these cell types during large network events such as sharp waves.

## Introduction

The hippocampus is one of the most studied brain regions in neuroscience. Since the study of patient H.M., the hippocampus has been established as an essential brain region for the formation of new memories (*Scoville and Milner, 1957*). In the decades following, a plethora of research has revealed insights into how the hippocampus contributes to memory processing. One hallmark of hippocampal activity that underlies memory formation is the sharp wave-ripple complex (SPW-R). SPW-Rs are large network events characterised by the synchronous discharge of huge numbers of hippocampal pyramidal (CA3) neurons (the sharp wave, SPW), followed by oscillatory activity downstream in the CA1 (ripple). During non-REM sleep, SPW-Rs essentially replay a compressed version of neural sequences that occurred during events preceding sleep (*Wilson and McNaughton, 1994*; *Lee and Wilson, 2002*). It is thought that these SPW-Rs play a key role in memory consolidation. Despite extensive research, the full cellular mechanisms underlying the initiation and propagation of SPW events are still not fully resolved.

The CA3 region in the hippocampus is considered the main generator of SPWs (*Buzsáki, 1986*). While inhibition is proposed to be instrumental in SPW generation, recent evidence has suggested that a subclass of pyramidal neurons may also play a pivotal role (*Hunt et al., 2018*). Diversity among pyramidal neuron populations is often overlooked when considering the role of these cells within neuronal circuits. Despite reports of variation within the pyramidal population in the hippocampus (*Bilkey and Schwartzkroin, 1990*; *Fitch et al., 1989*), much more attention has been paid to the heterogeneity of interneurons. However, several studies have reported functional and morphological heterogeneity within the pyramidal CA3 cell population (*Bilkey and Schwartzkroin, 1990*; *Sun et al., 2017*; *Marissal et al., 2012*; *Lee et al., 2015*). Attention is now turning to this rich assortment of pyramidal cells, and recent efforts have begun to tease apart the distinct roles of these sub-types in functional circuits (*Cembrowski and Spruston, 2019*; *Soltesz and Losonczy, 2018*; *Valero and de la Prida, 2018*). A recent study described two distinct sub-types of CA3 pyramidal neurons, differentiated by the presence or absence of complex spine structures called thorny excrescences (the postsynaptic site of input coming from mossy fibres of the dentate gyrus granule cells) (*Hunt et al., 2018*). The study showed that cells lacking these thorny excrescences, termed athorny pyramids, fire before thorny pyramids during SPWs (*Hunt et al., 2018*). Therefore, it is proposed that athorny cells play an important role in SPW initiation and, in turn, in memory processing in CA3. However, it is unknown how these two sub-types of pyramidal neuron are embedded in the local microcircuit. We have previously shown that CA3 pyramidal cells connect to each other at a high rate (8.8%) (*Sammons et al., 2024*). Here, we investigate the local sub-type specific connectivity between thorny and athorny CA3 pyramids and find a distinct asymmetry. When implementing this asymmetry into a computational model, we find that sub-type specific connectivity is crucial for the distinct firing times of athorny and thorny cells during SPWs.

## Results

To examine the connectivity between thorny and athorny pyramidal cells in CA3, we performed whole-cell patch clamp recordings from up to eight cells simultaneously. Cells were post hoc classified as thorny or athorny using biocytin labelling and confocal microscopy to determine the presence or absence of thorny excrescences (*Figure 1A*). In total, we recorded from 348 CA3 pyramids across the length of the CA3 (CA3a: 20 cells, CA3b: 274 cells, CA3c: 54 cells). Of these 348 pyramids, 229 were thorny and 119 were athorny (*Figure 1B*). We measured the distance from the soma to the first branch on the apical dendrite and found that thorny cells branched significantly closer to the soma than athorny cells (*Figure 1C*; median [IQR] for thorny: 12.5 [20.9] $\mu$m, athorny: 51.4 [38.0] $\mu$m; $p < 0.001$, Mann-Whitney-U test). Furthermore, we found that athorny cells tended to be located deeper in the pyramidal layer, towards the stratum oriens. Meanwhile, thorny cells were found throughout the deep-superficial axis of the pyramidal cell layer (*Figure 1D*, median [IQR] for thorny: 28 [32] $\mu$m, athorny: 12 [14] $\mu$m, $p < 0.001$, Mann-Whitney-U test). These results resemble findings from *Marissal et al., 2012* who observed similar differences in soma location and primary apical dendritic length between early and late born CA3 neurons, suggesting that thorny and athorny neurons may be developmentally distinct. We further found differences in maximum firing rate, rheobase and input resistance between thorny and athorny cells, but not in resting membrane potential (*Figure 1—figure supplement 1*).

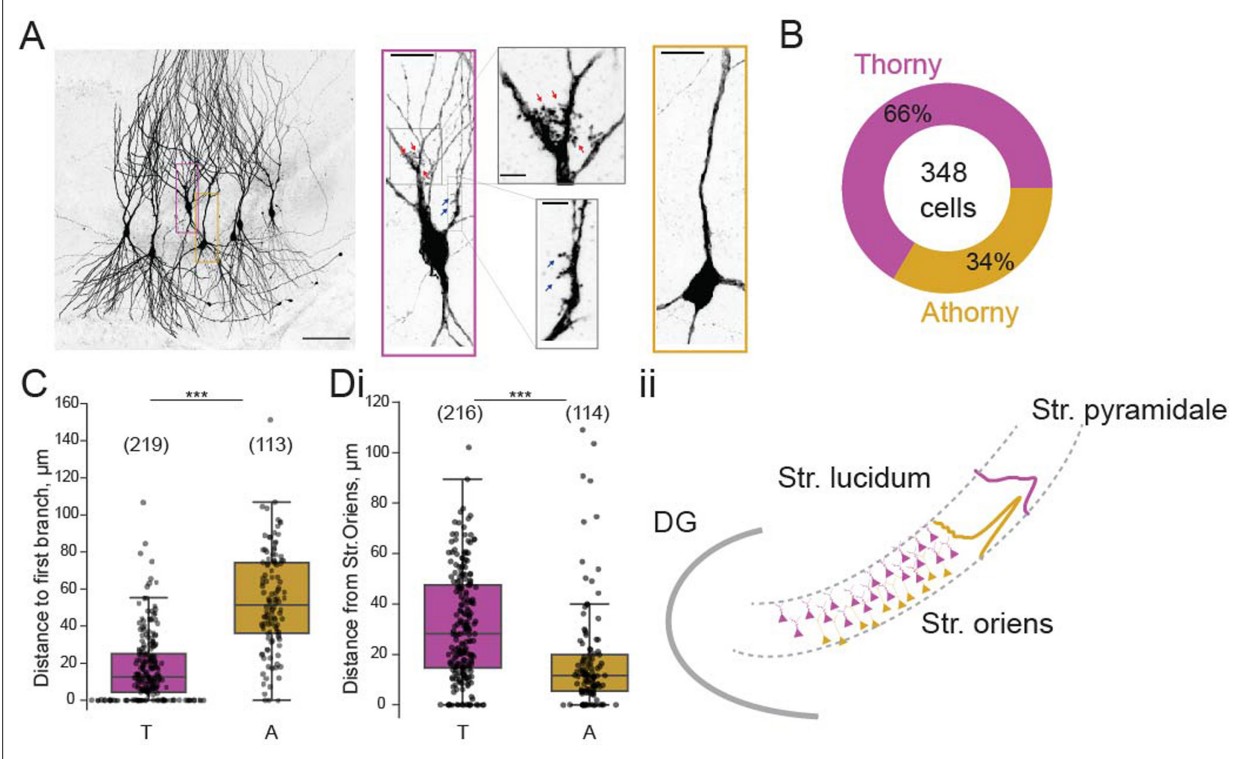

**Figure 1.** Proportion and distribution of thorny and athorny pyramidal neurons in CA3. (**A**) Left, image of seven pyramidal neurons recorded simultaneously and filled with biocytin to reveal thorny and athorny morphologies. Right, the magenta box contains a typical example of a thorny CA3 pyramid, gray boxes show close-ups of regions with thorns; yellow box shows a typical athorny pyramidal neuron. Scale bar in left image 100 *μm*, in magenta/yellow boxed insets 20 *μm*, in gray boxed insets 5 *μm*. (**B**) Proportion of thorny and athorny cells in total recorded pyramidal neurons. (**C**) Distance from soma to the first branch point for thorny (T) and athorny (A) CA3 pyramidal neurons. (**Di**) Location of thorny and athorny cell somata across the deep-superficial axis of the pyramidal layer. N numbers for each group shown above boxplots in parentheses. (**Dii**) Schematic depicting the distribution of thorny and athorny pyramids in the deep-superficial axis of the CA3 pyramidal layer.

The online version of this article includes the following figure supplement(s) for figure 1:

**Figure supplement 1.** Intrinsic properties of thorny (T) and athorny (A) cells.

Next, we looked at connection rates between these two pyramidal neuron populations. In our whole-cell patch clamp recordings, each cell was stimulated to elicit 4 action potentials, and post-synaptic traces were examined for potential synaptic coupling. We found a high rate of connectivity (15%) between athorny cells (*Figure 2Ai*), and from thorny onto athorny cells (11%; *Figure 2Aii*). Thorny cells connected to each other at a rate of 8% (*Figure 2Aiii*). Meanwhile, connections from athorny onto thorny cells were the least common, occurring at a rate of 4% (*Figure 2Aiv*). The overall rate of connectivity (65/734 = 8.9%) corresponds well to our previously reported high level of connectivity within the general CA3 pyramidal population (8.8%) (*Sammons et al., 2024*). We saw reciprocal connections between athorny-athorny pairs (2 reciprocally connected pairs; 4/92 connections = 4%) and thorny-thorny (2 reciprocally connected pairs; 4/362 connections = 1%) pairs of neurons. Synaptic connections were strongest amongst athorny-athorny cells, although no statistically significant differences were present across connection types (*Figure 2B*, median [IQR] amplitudes for athorny-athorny: 1.08 [0.56] mV; ii, thorny-athorny: 0.88 [1.03] mV; iii, thorny-thorny: 0.57 [0.55] mV; iv, athorny-thorny: 0.66 [0.25] mV; *p = 0.370*, Kruskal-Wallis test). EPSPs across all connection types had latencies below 3 ms (with the exception of a single connection between two athorny cells which had a latency of 3.58 ms) indicating that identified connections were monosynaptic (*Figure 2C*). We further looked at the failure rate of each synapse type. Athorny-athorny synapses had the lowest failure rate, although no statistical difference was observed between groups (*Figure 2D*, median [IQR] failure rate for athorny-athorny: 11.5 [20.5]%, thorny-athorny: 33.0 [36.2]%, thorny-thorny: 21.0 [36.3]%, athorny-thorny: 12.0 [47.5]%, *p = 0.729*, Kruskal-Wallis test). Additionally, we looked at synaptic dynamics to determine if synapse types had different plasticity qualities. Connections from thorny onto athorny neurons

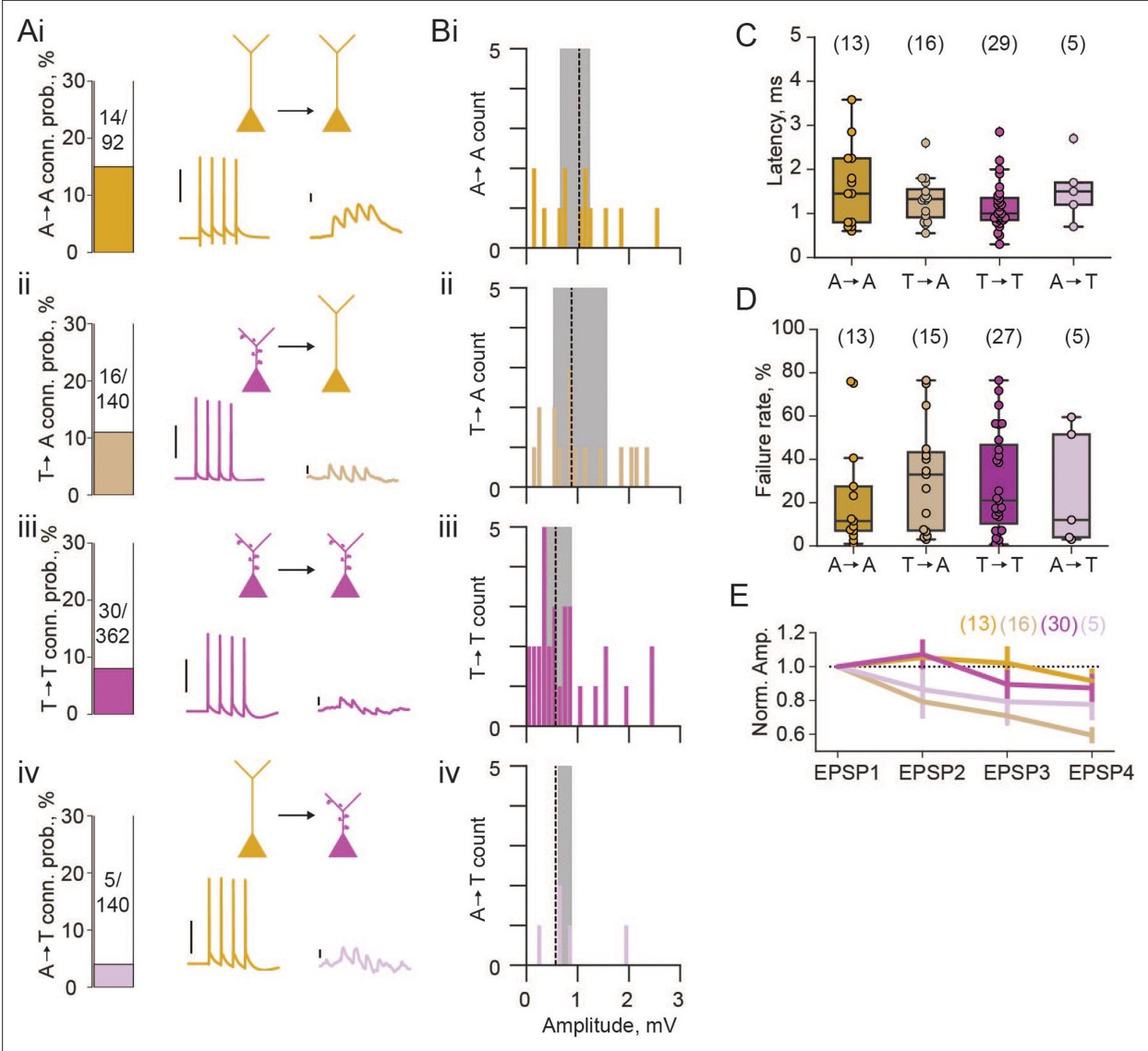

**Figure 2.** Properties of excitatory connections between athorny (**A**) and thorny (**T**) CA3 pyramids. (**A**) Connection probabilities (conn. prob.) and example connections: i, from A to A cells (A→A); ii, T→A; iii, T→T; iv, A→T. Scalebars for presynaptic action potentials, 40 mV; for postsynaptic responses, 0.5 mV. (**B**) Histograms of synaptic amplitudes of the different connection types. Dashed lines represent median values and shaded areas interquartile ranges. (**C**) Latency of different synaptic connection types, individual points show single connection values. (**D**) Failure rates of the different synaptic connection types. (**E**) Short-term plasticity dynamics of different synaptic connection types. Synaptic amplitudes are normalised to the first EPSP in the train of 4 and plotted as mean ± s.e.m. N numbers for groups shown above plots in parentheses.

showed significantly more synaptic depression than athorny-athorny connections (*Figure 2E*; p = 0.008, Kruskal-Wallis followed by Dunn's post hoc with Bonferroni correction; all other comparisons p > 0.05).

To determine the overall impact of each connection type within the local network, we calculated the synaptic product. This metric takes into account connection probability (*Figure 3Ai*), connection strength (*Figure 3Aii*), and size of the presynaptic population (*Figure 3Aiii*), thereby giving an estimate of how large the input onto the particular cell type is for any given presynaptic population. Thorny-athorny connections show the highest synaptic product, followed by athorny-athorny connections (*Figure 3Aiv*). Together, our results demonstrate a strong pattern of input onto athorny neurons and much weaker input onto thorny cells, particularly from the athorny sub-population (*Figure 3B*).

During SPW events, athorny (A) cells have been reported to fire before thorny (T) cells, suggesting that activity propagates in this direction (*Hunt et al., 2018*). Therefore, it might appear surprising that

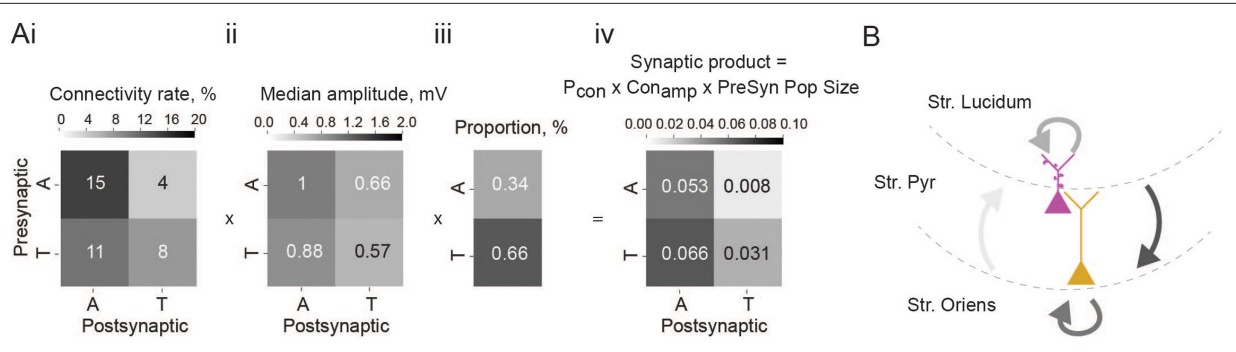

**Figure 3.** Summary of overall impact of each connection type. (**Ai**) Matrix showing connection rates between the four combinations of connection types, ii, matrix showing mean connection strength for the four connection types, iii, proportion of each cell type found in the CA3 pyramidal population, iv, matrix showing the synaptic product, calculated as the product of the matrices in i and ii multiplied by the presynaptic population size shown in iii. (**B**) Schematic depicting the connections between the two pyramidal cell types in the CA3, line colour is coded by connection impact.

both connectivity and synaptic product present the opposite asymmetry, with low values for athorny-thorny (A→T) and high values for thorny-athorny (T→A) connections. To understand what dynamics can be expected based on this microcircuit architecture, we constructed a model network in which T and A neurons are connected according to the experimentally observed connections (*Figure 4A*). In addition to the two pyramidal cell populations, we included two classes of interneurons that have been suggested to play fundamental roles in SPW generation: PV⁺ basket cells (B), which are active during SPWs, and a putative class of anti-SPW interneurons (C), which fire outside SPWs, keeping the other populations inhibited (*Evangelista et al., 2020*). Strong reciprocal coupling between the two inhibitory populations gives rise to a bistability between a SPW state and a non-SPW state, and the network alternates between these two states due to adaptation in pyramidal cells (*Figure 4Bi*), as proposed by *Levenstein et al., 2019* as the driving mechanism of SPWs. We tuned model parameters (see Materials and methods) such that the SPW event incidence is ≈1/s (with stochastic onsets driven by finite-size fluctuations) and the average event duration is ≈80 ms (*Figure 4Bi*).

In our simulations, a SPW event starts when B cells suppress enough C cells to disinhibit pyramidal neurons. Among the two pyramidal cell populations, A cells subsequently emit a larger number of early spikes, due to a lower rheobase (documented by *Hunt et al., 2018*, *Linaro et al., 2022*, and our own data (*Figure 1—figure supplement 1*)) and a steeper f-I curve (*Hunt et al., 2018*; *Linaro et al., 2022*). These initial spikes recruit many further A cells, due to the high A→A connectivity, but only a few T cells, due to the low A→T connectivity. On the other hand, A cells drive the growth of B cells, which in turn inhibit T cells (*Figure 4Bi–ii*): as a result, A cells have a net *inhibitory* effect on T cells, which get initially suppressed and can only start firing when the activity of A cells decreases, due to a surge in adaptation (*Figure 4Biii*). Because T cells are also adaptive, their firing rate also reaches a peak and decreases, ending the SPW event. Together, these dynamics result in clearly distinct peaks of A and T population activities, with an average delay of 29 ms between the peaks (*Figure 4Bi–ii*). This long delay matches the data by *Hunt et al., 2018*, which would be hard to explain if T cells were directly recruited by A cells, with monosynaptic latencies shorter than three milliseconds (*Figure 2C*).

A key ingredient for these dynamics is that the proportion between the A→T and A→A connectivity is such that A has a net inhibitory effect on T, resulting in a long delay between the peaks. We confirm this intuition by varying each of the four connectivities in the model and find that a decreased A→A (*Figure 4Ci*) or increased A→T (*Figure 4Cii*) connectivity would indeed prevent the initial suppression of T cells, as activity would build up together in both populations, resulting in almost simultaneous peaks. On the contrary, further increasing A→A or decreasing A→T would more strongly suppress T cells, which could only fire after most A cells have adapted and fallen silent, with delays even over 100 ms. For particularly low A→T connectivities, we even observe two separate A peaks, since the delay becomes so long that A cells partially recover from adaptation by the end of the SPW (*Figure 4Cii, left inset*). Analogously, connections from T neurons determine whether these provide net excitation or inhibition to A neurons. However, albeit affecting the relative size of the A and T peaks, such connections on their own cannot prevent the early activation of the A population, which depends on single-neuron parameters. Therefore, connections from T cells play a role only in the second part

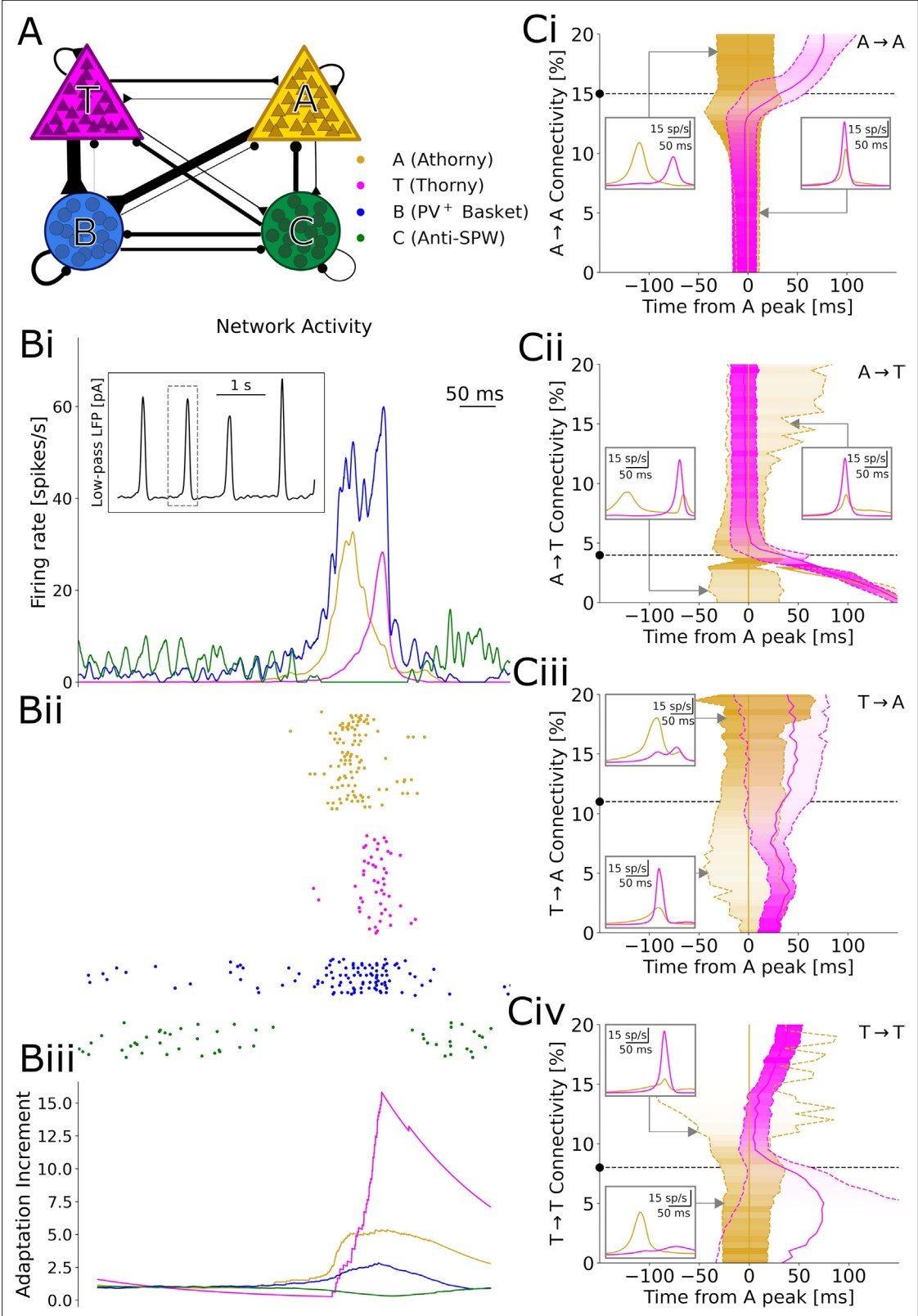

**Figure 4.** Results of numerical simulations. (**A**) Network scheme. (**Bi**) Firing rates before, during, and after a sharp wave-ripple complex (SPW). Inset: low-pass filtered estimate of the LFP over a longer window of 10 s. ii, Spike raster plot of a representative sample of each neuron type. iii, Relative increment of the average adaptive currents received by each population with respect to a 200 ms baseline before the event. (**C**) Effects of varying each connectivity from its default value, marked by a black dashed line and dot. Continuous gold/magenta lines indicate the peak time of each population rate (with

*Figure 4 continued on next page*

*Figure 4 continued*

the peak of A always plotted at 0), while dashed ones represent the time at which the rate equals 25% of the respective peak. The peak size for each connectivity value is color-coded. In these simulations, the synaptic strengths of all the excitatory-to-excitatory synapses were up- or down-scaled by a common factor, in order to obtain SPW events with a comparable size (measured based on the activity of B interneurons, see Materials and Methods). Insets: firing trace of each population averaged over many events, for particular connectivity values highlighted by the gray arrows.

The online version of this article includes the following figure supplement(s) for figure 4:

**Figure supplement 1.** Onset f-I curves for each neuron type, calculated, for comparability, by delivering a constant current for 500 ms, like in *Hunt et al., 2018*.

**Figure supplement 2.** Simulations with short-term synaptic depression.

**Figure supplement 3.** Simulations with heterogeneity.

of an event (*Figure 4Ciii and Civ*). The model dynamics and the effects of excitatory-to-excitatory connectivities qualitatively remain unaltered if we include in the model the experimentally observed short-term depression and variability of synapses (*Figure 4—figure supplements 2 and 3*).

To test whether the described relationship between connectivity and delay holds across the parameter space, we explore the six possible combinations of the connectivities examined so far. We find that long delays are consistently found when A→A is large (*Figure 5A1,2,3*) and when A→T is small (*Figure 5A3,4,5*), provided that delays do not become so long that population T does not activate at all. Specifically, varying A→T and T→A together offers a good overview of the possible combinations

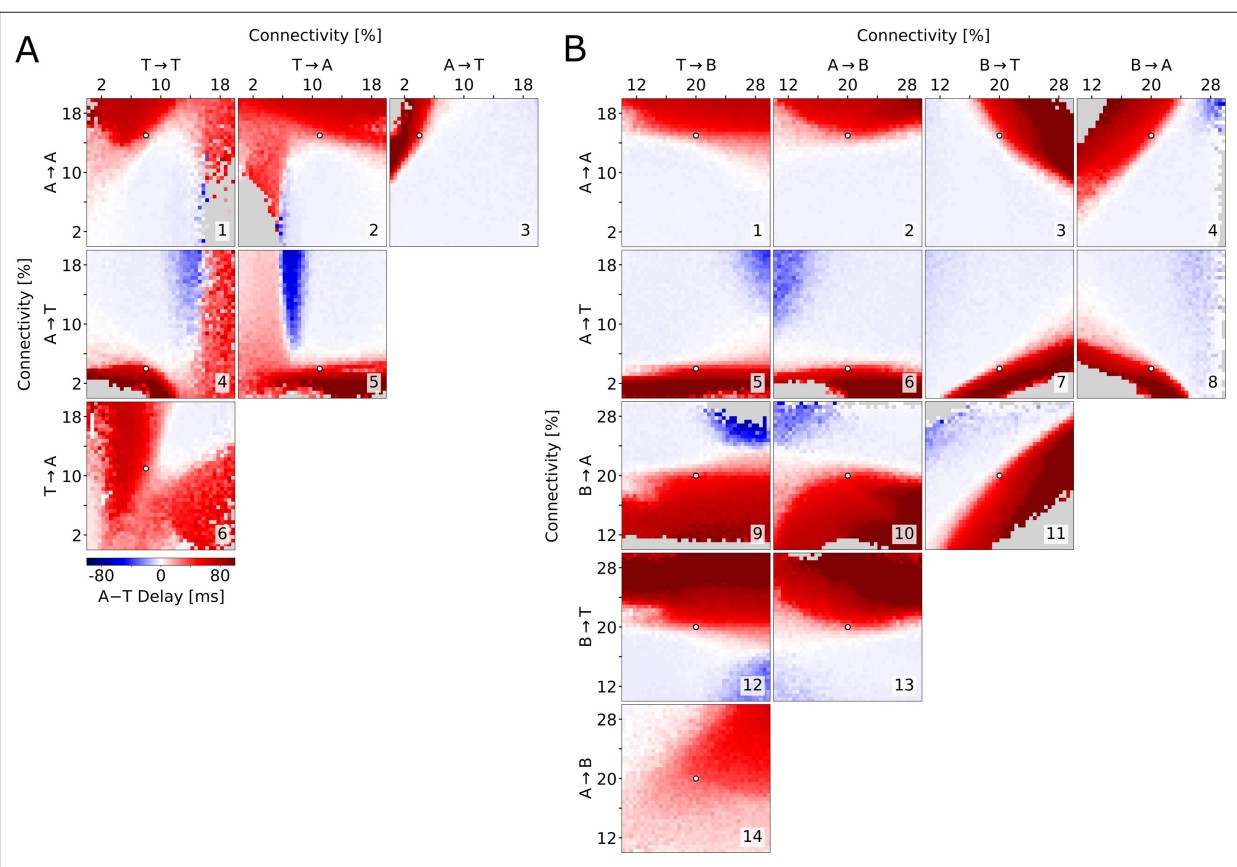

**Figure 5.** Exploration of the parameter space. Average delay between the athorny and the thorny peak, as a function of two connectivity parameters (pairs of excitatory-to-excitatory connections in (**A**), pairs involving inhibitory connections in (**B**)). In each simulation, the connectivity rates were varied from their default value (marked by a white dot), while all the excitatory-to-excitatory synaptic strengths were scaled, as in *Figure 4*, in order to obtain sharp waves with a comparable size. Data points are plotted only if both populations exceed a lower threshold on the firing rate; otherwise, they are grayed out. The same simulations as in (**A**), but with a different scaling, are reported in *Figure 5—figure supplement 1*.

The online version of this article includes the following figure supplement(s) for figure 5:

**Figure supplement 1.** Alternative scaling.

of cooperative and competitive interactions between the two populations (*Figure 5A5*). If these connectivities were both high, then there would be no competition and the populations would reach their peaks at approximately the same time (top right corner). Conversely, if both connectivities were low, either population could potentially suppress the other. This winner-takes-all competition would be won by the athorny cells because of their earlier activation due to their intrinsic properties (bottom left corner). When A→T is low and T→A is high (bottom right corner), matching experimental conditions, only A can suppress T, resulting in the long delay, while when A→T is high and T→A is low, A is suppressed by T, after nevertheless displaying some early activity. Afterwards, there may or may not be a later peak for the A neurons, after T neurons have adapted. This is why, in the top left corner of *Figure 5A5*, there can either be a positive or long negative delay, depending on whether the first (small) or second (absent, for some parameters) A peak is taken into account.

The arguments presented so far assume the existence of indirect inhibitory pathways through the B interneurons, which become dominant in case the direct excitatory pathways are weak. How strong do these pathways need to be for T neurons to be initially suppressed, causing two distinct peaks to emerge? To address this question, we varied pairs of connectivity rates to and from B interneurons (B→A, B→T, A→B, and T→B — *Figure 5B, 9-14*) and combinations thereof with the relevant connections among excitatory neurons (A→A in *Figure 5B, 1-4* and A→T in *Figure 5B, 5-8*). In this analysis, it clearly emerges that connections *to* the B interneurons have only a minor effect on the delay, while connections *from* the B interneurons are as important as the excitatory connections outlined above. Although the actual values of the inhibitory connectivities are unknown, these simulations demonstrate that there is a broad, clearly distinct, region of the parameter space that supports a long delay between the peaks of A and T. In addition, we see that B interneurons optimally contribute to the suppression of T when they primarily target T (*Figure 5B, fourth row and third column*) rather than A (*Figure 5B, third row and fourth column*). Interestingly, in the examples in *Figure 4* the B→T connections are weaker than B→A ones (total connection strengths are displayed graphically in *Figure 4A*). The fact that the long delay nevertheless emerges is due to the measured asymmetric connectivity between the two excitatory populations, in particular the low connectivity from A to T neurons. In conclusion, modeling shows that not only can T cells activate after A cells even if the A→T connectivity is low, but such a low connectivity is also crucial to explain the delay seen in the data by *Hunt et al., 2018* and in our model.

## Discussion

Our work combines electrophysiology and computational modeling to determine connectivity patterns within the local CA3 excitatory network and to show how these rules could govern the timing of excitatory subpopulation activation during sharp wave events.

We corroborate previous findings that a subset of CA3 pyramidal cells appears to lack thorny excrescences typical of mossy fiber inputs, and thus, the population can be divided into thorny and athorny neurons (*Hunt et al., 2018*). Moreover, we find that the connectivity between these classes is distinctly asymmetric, with athorny cells receiving ample input from themselves and thorny cells, but thorny cells receiving sparse inputs from athorny pyramids. Though classified by location in the pyramidal layer (deep and superficial) rather than thorny or athorny, two other groups have recently reported similar findings using two different experimental methods (*Watson et al., 2024*; *Layous et al., 2025*). Both studies report superficial neurons frequently innervating themselves and deep CA3 pyramids and much less frequent connectivity from deep to superficial cells. Together with our data showing that athorny cells tend to be located deeper in the pyramidal layer, these results reflect a similar pattern of asymmetric connectivity.

Our modeling shows that asymmetric connectivity is crucial for SPW events to have two distinct peaks with only partially overlapping activity for the two pyramidal populations, matching the data by *Hunt et al., 2018*. The long delay (tens of milliseconds) in our model is explained by the ambivalent role of athorny neurons, which on the one hand switch the network to the SPW state, but on the other hand initially disynaptically suppress the thorny neurons. The switch to the SPW state occurs because PV[+] basket cells take over from anti-SPW interneurons, a putative class proposed by *Evangelista et al., 2020* to explain the paradoxical triggering of SPWs by in vitro stimulation of PV[+] basket cells (*Schlingloff et al., 2014*). Although their existence in CA3 still needs to be demonstrated, an interneuron class with similar properties was identified in CA1 (*Dudok et al., 2021*). During a SPW,

the initial suppression of thorny neurons occurs through the shared pool of PV⁺ basket cells. The long delay that we observed in our model does not require special asymmetries in the connectivity between the interneurons and the pyramidal populations, given the measured values of excitatory connectivity. However, at least some baseline connectivity from athorny to basket and from basket to thorny neurons is required for the proposed disynaptic inhibition to exist. More detailed data on the interneuron connectivity would further deepen our understanding of subpopulation timing and other aspects of the initiation and propagation of SPW-ripple complexes.

The further functional relevance of these two cell populations remains unclear. The CA3 pyramidal population has two developmentally distinct groups — early-born and late-born cells (*Marissal et al., 2012*). These developmentally distinct subpopulations have been shown to exhibit functionally different roles in memory encoding, with late-born cells recruited earlier in the encoding process than early-born cells (*Kveim et al., 2024*). Currently, it is not known whether thorny and athorny cells might map onto the two developmentally distinct populations. However, certain features of early- and late-born neurons are suggestive of an overlap with athorny and thorny cells, respectively. Early-born neurons tend to be located closer to the stratum oriens and have longer primary apical dendrites (*Marissal et al., 2012*), reminiscent of our findings on athorny cells. Moreover, modulation of thorn density has been reported through excessive training in spatial memory tasks and chemically via steroid hormones (*Gómez-Padilla et al., 2020*; *Hatanaka et al., 2009*; *Tsurugizawa et al., 2005*). However, it is unclear whether modulation may result in the complete transition of thorny neurons to athorny cells or growth of thorns in previously athorny neurons.

The present work extends our existing knowledge of how the local excitatory network in the CA3 region is organized. In line with recent evidence that the CA3 pyramidal population is non-homogeneous, we find that connectivity between these pyramids is asymmetric across subpopulations. Our modeling work shows that this asymmetry is key in maintaining the separation of peak firing between the two populations, thorny and athorny, during sharp wave events.

# Materials and methods
## Electrophysiology
### Slice preparation
Mice (C57/Bl6n, bred in-house and originating from Charles River, P25+, average age: P40, both sexes) were decapitated following isoflurane anesthesia. Brains were removed and transferred to ice-cold, sucrose-based artificial cerebrospinal fluid (sACSF) containing (in mM) 50 NaCl, 150 sucrose, 25 NaHCO₃, 2.5 KCl, 1 NaH₂PO₄, 0.5 CaCl₂, 7.0 MgCl₂, 10 glucose, saturated with 95% O₂, 5% CO₂, pH 7.4. Slices (400 $\mu$m) were cut in a horizontal plane on a vibratome (VT1200S; Leica) and stored in an interface chamber at 32–34 °C. Slices were perfused at a rate of ~1 ml/min with artificial cerebrospinal fluid (ACSF) containing (in mM) 119 NaCl, 26 NaHCO₃, 10 glucose, 2.5 KCl, 2.5 CaCl₂, 1.3 MgCl₂, 1 NaH₂PO₄, and continuously oxygenated with carbogen. Slices were allowed to recover for at least 1.5 hr after preparation before they were transferred into the recording chamber.

### Connectivity
Recordings were performed in ACSF at 32–34°C in a submerged-type recording chamber. Cells in the CA3 were identified using infrared differential contrast microscopy (BX51WI, Olympus). We performed somatic whole-cell patch-clamp recordings (pipette resistance 3–5 MΩ) of up to eight cells simultaneously. One cell was stimulated with a train of four action potentials at 20 Hz, elicited by 2–3 ms long current injections of 1.5–4 nA. For characterization to confirm the targeting of pyramidal cells, increasing steps of current were injected (1 s, increment: 50 pA). In some cells, a hyperpolarizing or depolarizing holding current was applied to keep the membrane potential at -60 mV. The intracellular solution contained (in mM) 135 potassium-gluconate, 6.0 KCl, 2.0 MgCl₂, 0.2 EGTA, 5.0 Na₂-phosphocreatine, 2.0 Na₂-ATP, 0.5 Na₂-GTP, 10 HEPES buffer, and 0.2% biocytin. The pH was adjusted to 7.2 with KOH. Recordings were performed using Multiclamp 700B amplifiers (Molecular Devices). Signals were filtered at 6 kHz, sampled at 20 kHz and digitized at 16 bit resolution using the Digidata 1550 and pClamp 10.7 (Molecular Devices). A subset of the data (n = 238 out of 348 cells) was published in a separate study (*Sammons et al., 2024*).

## Data analysis — connectivity

Cells with a membrane potential less negative than −50 mV and a series resistance higher than 30 MΩ were discarded. The connectivity screen underwent a quality control step such that only sweeps were kept if presynaptic action potentials reversed above 0 mV and the membrane potential did not deviate by more than 10% within a sweep or with reference to the first sweep. Synaptic connections were identified when there was a postsynaptic potential corresponding to presynaptic stimulation in the averaged trace of 40–50 sweeps. A baseline period (2 ms) just prior to stimulation and the averaged postsynaptic peak during the first action potential was used for the analysis of EPSP amplitudes and synaptic delays. Only pairs in which the first postsynaptic peak was clearly discernible were used for analysis. To analyze short-term plasticity dynamics, postsynaptic traces were deconvolved as described by *Richardson and Silberberg, 2008*. The time constant, $\tau$, was set to 55 ms and the deconvolved trace was low-pass filtered. Subsequent evoked EPSP peaks were normalized to the first evoked EPSP in the trace. Synaptic dynamics were compared across connection types by comparing the ratio of the first and fourth EPSPs across groups. Failure rate was calculated by dividing the number of sweeps in which an EPSP was observed by the total number of sweeps. This value was calculated for each of the possible four EPSPs corresponding to the four presynaptic action potentials, and then a total sum for each cell was taken. For all boxplots, boxes cover quartiles and whiskers show entire distribution of data excluding outliers, which are shown additionally as filled black circles and considered to be 1.5 x interquartile range. In *Figure 2C and D* all data points are shown as coloured, filled circles. Statistics were carried out in Python using the scipy stats module, with a significance level set to 0.05. Data were first checked for normality using the Shapiro-Wilk test. Subsequently, non-parametric tests were performed as appropriate and the Bonferroni correction method was applied to account for multiple comparisons. Raw data used to create *Figures 1–3* is available on FigShare at https://doi.org/10.6084/m9.figshare.29390549.v1.

## Data analysis — immunohistochemistry and neuroanatomy of principal cells

After recording, slices were transferred into a fixative solution containing 4% paraformaldehyde in 0.1 M phosphate buffer. Biocytin labelling was revealed by incubating slices in streptavidin conjugated to Alexa 488 (diluted 1:500) overnight in a solution of PBS containing 2.5% normal goat serum and 1% Triton. The slices were then mounted in Mowiol (Sigma-Aldrich). Image stacks of specimens were imaged on an Olympus BX61 FV1000 confocal microscope. Images were taken using a ×20 objective with a pixel size of 0.62 μm and a z-step size of 1 μm. The morphology of the pyramidal neurons was scored as 'thorny' or 'athorny' based on the presence or absence of thorny excrescences, respectively. Each cell was scored by at least three independent investigators to ensure that in ambiguous cases a consensus was reached. Location of cells relative to the stratum oriens was measured in Fiji (*Schindelin et al., 2012*) using the line tool and drawing a perpendicular line from the base of the cell soma to the estimated edge of the pyramidal layer at the side of the stratum oriens.

## Computational model

### Model equations

Neurons are modeled as adaptive exponential (AdEx) integrate-and-fire neurons (*Brette and Gerstner, 2005*). This level of complexity (two dynamic variables: voltage and adaptation) is necessary to capture the diverse firing patterns of different neural populations. In addition, neuronal adaptation has been proposed as the main mechanism governing the alternation between SPW and non-SPW states (*Levenstein et al., 2019*). In the AdEx model, the membrane potential $V_i$ of each neuron $i$ evolves according to the equation

$$C\dot{V}_i(t) = -g_L(V_i(t) - E_L) + g_L\Delta_T \exp\left(\frac{V_i(t) - V_T}{\Delta_T}\right) - u_i(t) + I_{ext} + I_{syn}(t) \tag{1}$$

where $C$ is the membrane capacitance, $E_L$ is the resting potential, $g_L$ is the leak conductance, and $V_T$ is the threshold potential. Slightly above this threshold, the membrane potential escapes from the basin of attraction of $E_L$ and begins an exponential upswing with a slope $\Delta_T$. As soon as the upswing reaches a conventional value $V_{stop}$, a spike is emitted and $V_i$ is reset to a value $V_{reset}$ and fixed there for

a refractory time $\tau_{ref}$. Neurons receive an internal feedback inhibition $u_i(t)$, representing an adaptive current, which evolves according to

$$\tau_u \dot{u}_i(t) = -u_i(t) + a(V_i(t) - E_L) \tag{2}$$

in which $a$ is the voltage-coupling of adaptation and $\tau_u$ is its timescale. Upon spiking, $u$ is increased by an amount $b$ (spike-triggered adaptation). Neurons receive a constant external input $I_{ext}$ and a synaptic current $I_{syn}(t) = \sum_J g_i^J(t)(V_i(t) - E_{rev}^J)$, where $E_{rev}^J$ is the reversal potential for the neurotransmitter used by the pre-synaptic population $J$, and $g_i^J(t)$ is the total synaptic conductance received from the neurons in population $J$, which obeys

$$\dot{g}_i^J(t) = -\frac{g_i^J}{\tau_d^J} + \sum_{f,j} \delta(t - t_j^f - \tau_l)p_{IJ}w_{IJ}, \tag{3}$$

where $\tau_d^J$ is the synaptic decay constant for population $J$, and $\tau_l$ is the synaptic latency. The contribution of each pre-synaptic spike at time $t_j^f$ is determined by a connection probability $p_{IJ} \in [0, 1]$ and a weight $w_{IJ}$. In the model variant with short-term synaptic depression, for each pair of connected excitatory neurons $i$ and $j$ in populations $I$ and $J$, the weight $w_{IJ}$ is scaled by an efficacy factor $e_{ij}$, which follows its own dynamics:

$$\dot{e}_{ij} = \frac{1 - e_{ij}}{\tau_{dep}} - \sum_f \delta(t - t_j^f)e_{ij}\eta_{dep}, \tag{4}$$

where $\tau_{dep}$ is the time constant of synaptic depression and $\eta_{dep}$ is the depression rate. In the model variant with heterogeneous parameters, the synaptic weights $w_{ij}$ between excitatory neurons $j$ and $i$ are sampled from a distribution with mean $w_{IJ}$ and standard deviation $\frac{1}{4}w_{IJ}$. The synaptic latencies between excitatory neurons are sampled from a distribution with mean $\tau_l$ and standard deviation $\frac{1}{4}\tau_l$, while the resting potential of excitatory neurons has standard deviation $\frac{1}{20}E_L$ and their leak conductance has standard deviation $\frac{1}{4}E_L$.

## Single neuron parameters

We consider four different neural populations: thorny pyramids (T), athorny pyramids (A), PV⁺-basket cells (B), and anti-SPW interneurons (C). The latter are modeled as CCK+-basket cells. For each population, parameters were chosen in order to be close to the single-neuron physiology. For A and T neurons, we follow the main figures and supplementary data by *Hunt et al., 2018* and *Linaro et al., 2022*, since they performed detailed single-neuron physiological characterization of the two neuron types. Namely, athorny neurons were shown to have a higher input resistance, a higher resting potential, and a lower firing threshold than their thorny counterparts, and both kinds have a high reset potential. In particular, we reset athorny neurons above the threshold, because this is how the AdEx model produces bursting (*Naud et al., 2008*), a feature that has been reported in this cell type (*Hunt et al., 2018*). Our parameters result in a lower rheobase for athorny than for thorny neurons (*Hunt et al., 2018*, *Linaro et al., 2022*, *Figure 1—figure supplement 1*). Interneuron parameters were based on data from CA3, if available (*Fidzinski et al., 2015*; *Pelkey et al., 2017*), or otherwise from other hippocampal subfields (*Ledri et al., 2012*; *Pawelzik et al., 2002*; *Tricoire et al., 2011*).

The parameters of adaptation cannot be directly compared to physiological values, because this variable summarizes a multitude of different currents, each with its own size and timescale (*Benda, 2021*). Therefore, we firstly aimed at reproducing the f-I curves of different neurons, when available (*Figure 4—figure supplement 1*). Thorny and athorny f-I curves were compared to those measured by *Linaro et al., 2022*, while for PV⁺- basket cells we used CA3 data from *Fidzinski et al., 2015*. In addition, the large spike-coupling $b$ and long timescale $\tau_u$ of pyramidal adaptation allow to reproduce the strong firing rate accommodation typical of these cells (*Storm, 1990*; *Hunt et al., 2018*), while these parameters are smaller in A and especially B cells, which can sustain a high firing rate without significant accommodation (*Pelkey et al., 2017*). In the AdEx model, if the voltage-coupling $a$ is strong enough, spiking happens through a Hopf bifurcation, which is responsible for phenomena like transient spiking and class 2 behaviour (*Touboul and Brette, 2008*). Therefore, we set this parameter to 0 for thorny cells, in which these behaviours are absent, and to a higher value for athorny cells, which seem to exhibit transient spiking for intermediate values of a constant input (*Hunt et al., 2018*),

**Table 1.** Single neuron parameters.

| | | Athorny (A) | Thorny (T) | PV$^+$- Basket (B) | Anti-SPW (C) |
|---|---|---|---|---|---|
| Population size | | 2700 | 5300 | 150 | 100 |
| $C$ | [pF] | 200 | 200 | 100 | 100 |
| $g_L$ | [nS] | 8 | 11 | 8 | 5 |
| $E_L$ | [mV] | –60 | –70 | –55 | –57 |
| $V_{thr}$ | [mV] | –48 | –44 | –40 | –40 |
| $V_{reset}$ | [mV] | –42 | –46 | –57 | –52 |
| $a$ | [nS] | 4 | 0 | 6 | 2.5 |
| $b$ | [pA] | 85 | 150 | 25 | 20 |
| $\tau_u$ | [ms] | 200 | 200 | 50 | 100 |
| $\tau_{ref}$ | [ms] | 3 | 3 | 3 | 3 |
| $\Delta_T$ | [mV] | 2.5 | 2.5 | 2.5 | 2.5 |
| $I_{ext}$ | [pA] | 140 | 285 | 180 | 160 |

and for interneurons. In particular, for B cells, we could reproduce the discontinuity around 15 Hz typical of fast-spiking interneurons (*Gerstner et al., 2014*). Neuronal parameters and their values are summarized in *Table 1*.

## Network parameters

Each population size is based on an estimation of its representation in a 400-$\mu m$-thick CA3 slice, according to the quantitative assessment by *Bezaire and Soltesz, 2013*. Pyramidal neurons are divided into thorny and athorny according to the 66–34% ratio that we determined experimentally. The background currents $I_{ext}$ are constant and correspond to the non-transient rheobase $\rho$, plus 10%, with the exception of population A, which receives +40% because it is responsible for keeping the other neurons inhibited for most of the time. This assumption is reasonable, since CCK$^+$- basket cells 'receive a far less efficient local excitatory drive, but are exposed to modulatory effects of extrinsic inputs (*Freund, 2003*).'

Neurons are connected to each other with a probability $p_{IJ}$, depending on the pre- and post-synaptic population $J$ and $I$. For excitatory-to-excitatory connections, these probabilities have the values that we assessed experimentally. For the other connections, the existent literature is too inconsistent to derive coherent conclusions (*Gulyás et al., 1993*; *Maccaferri et al., 2000*; *Mátyás et al., 2004*; *Bezaire and Soltesz, 2013*; *Campanac et al., 2013*; *Kohus et al., 2016*; *Pelkey et al., 2017*; *Dudok et al., 2021*): therefore, in order to minimize the number of assumptions not based on solid evidence, they were all given the same probability 0.2.

Excitatory-to-excitatory synaptic weights were all set to 0.2 nS, since differences in EPSP sizes were not found to be significant (*Figure 2C*). These values correspond to an EPSP size of 0.1 mV, which is lower than the ones measured experimentally, but compensates for the fact that they directly affect the (somatic) membrane potential of the post-synaptic neurons neurons and that connections are homogeneous. The weights involving population C were chosen in order to satisfy the basic requirements for bistability and disinhibition dynamics: the search for the bistable region of the parameter space was guided by the insights previously obtained in the bifurcation analysis of a three-population model of CA3 (*Evangelista et al., 2020*). Although our model has one more population, we found that the basic requirements are the same: pyramidal cells need to more strongly excite interneurons B and to be more strongly inhibited by interneurons C. In addition, populations C and B need to have strong inhibitory couplings between each other. For firing rate requirements, we assumed, following *Evangelista et al., 2020*, that C neurons fire ~10 spikes/s in non-SPW states and are almost silent during SPWs. These choices are assumptions on population C, which still need to be tested experimentally. For the connections between B neurons and pyramidal cells, the main criterion was to balance the

**Table 2.** Network parameters.

Parameter adjustments for the model variants with short-term synaptic depression and heterogeneity are reported in *Supplementary file 1*.

| | | From A | From T | From B | From C |
|---|---|---|---|---|---|
| $p_{AI}$ | | 15% | 11% | 20% | 20% |
| $p_{TI}$ | | 4% | 8% | 20% | 20% |
| $p_{BI}$ | | 20% | 20% | 20% | 20% |
| $p_{CI}$ | | 20% | 20% | 20% | 20% |
| $w_{AI}$ | [nS] | 0.2 | 0.2 | 2.15 | 15 |
| $w_{TI}$ | [nS] | 0.2 | 0.2 | 0.8 | 15 |
| $w_{BI}$ | [nS] | 0.7 | 0.5 | 6 | 9 |
| $w_{CI}$ | [nS] | 0.1 | 0.05 | 5 | 3 |
| $\tau_d$ | [ms] | 2 | 2 | 4 | 4 |
| $E_{rev}$ | [mV] | 0 | 0 | –70 | –70 |
| $\tau_l$ | [ms] | 1 | 1 | 1 | 1 |

effect of the strong recurrent excitation, in order to achieve a realistic firing rate for pyramidal neurons and B cells themselves during SPWs. For B neurons, we based on estimates on 5–10 spikes/s in the non-SPW periods and fast spiking at 50–70 during SPWs (*Klausberger and Somogyi, 2008*; *Lapray et al., 2012*; *Varga et al., 2012*; *Hájos et al., 2013*). Pyramidal neurons are almost silent (0–1 spikes/s) in non-SPW periods and fire on average 10–20 spikes/s in SPW events (*Klausberger and Somogyi, 2008*; *Lapray et al., 2012*; *Hájos et al., 2013*; *English et al., 2014*). In order to satisfy these requirements, inhibitory weights needed to be about one order of magnitude larger than excitatory ones, which is partially in accordance with the hippocampal interneuron literature mentioned above, and partially necessary because not all kinds of interneurons are included in the network. The resulting non-SPW activity of our pyramidal populations is 0.4 spikes/s for T cells and 0.4 spikes/s for A cells.

Regarding the other synaptic parameters, all the latencies were set to 1 ms, glutamatergic and GABAergic reversal potentials have the typical values of 0 mV and –70 mV, respectively, and the former are assumed to be twice as fast as the latter (*Geiger et al., 1995*; *Bartos et al., 2002*). Network parameters are summarized in *Table 2*.

## Network activity

SPW events are identified based on the current flowing from B cells to the excitatory ones, which is thought to represent most of the LFP signal observed in the *stratum pyramidale*. This signal is low-pass filtered up to 5 Hz, in order to cover the whole duration of an event. In this signal, peaks higher than 50 pA are regarded as SPWs. The beginning and end of the events are defined as the times at which the low-pass-filtered LFP crosses the value $\frac{1}{2}(L_p - L_0)$, where $L_p$ is the LFP peak of each event and $L_0$ is a baseline value. Within each sharp wave, we record the peak timing of the firing rate of both the athorny and the thorny populations. In case a single population presents two peaks that are more than 50 ms apart, they are recorded as separate peaks and only the first one is used for the statistics in *Figure 5* and *Figure 5—figure supplement 1*.

When varying connectivities as in *Figures 4 and 5*, we up- or down-scale the synaptic strength $w_{IJ}$ of all the four excitatory-to-excitatory connections by a common factor, in order to still obtain sharp waves with a similar average size of the LFP peak (±10 pA). Since the LFP peak is estimated based on the current flowing from the B interneurons, this criterion is not biased toward either pyramidal population. In *Figure 5—figure supplement 1*, we repeat the simulations in *Figure 5A*, but this time the scaling is done with the synaptic strength of the A→B and T→B connections, which are highly relevant for the sharp wave size, but in *Figure 5B* were found to be little relevant for the timing of the peaks.

## Acknowledgements

The authors would like to thank Susanne Rieckmann and Anke Schönherr for excellent technical assistance, Linda Brenndörfer for help with confocal microscopy and Antje Fortströer for administrative assistance. Funding sources: German Research Foundation: project 327654276 – SFB 1315 (DS, RK), project 184695641 – SFB 958 (DS), project 431572356 (DS); Germany's Excellence Strategy – Exc-2049–390688087 (NeuroCure to DS, RPS and MO), project 503954250 (MO); European Research Council Horizon 2020 grant 810580 – BrainPlay (DS); Federal Ministry of Education and Research project 01GQ1420B – SmartAge (DS).

## Additional information

### Funding

| Funder | Grant reference number | Author |
|---|---|---|
| Deutsche Forschungsgemeinschaft | 327654276 | Richard Kempter Dietmar Schmitz |
| Deutsche Forschungsgemeinschaft | 184695641 | Dietmar Schmitz |
| Deutsche Forschungsgemeinschaft | 503954250 | Marta Orlando |
| Deutsche Forschungsgemeinschaft | 431572356 | Dietmar Schmitz |
| Deutsche Forschungsgemeinschaft | Exc-2049-390688087 | Marta Orlando Dietmar Schmitz Rosanna P Sammons |
| European Research Council | 10.3030/810580 | Dietmar Schmitz |
| Bundesministerium für Bildung und Forschung | 01GQ1420B | Dietmar Schmitz |

The funders had no role in study design, data collection and interpretation, or the decision to submit the work for publication.

### Author contributions

Rosanna P Sammons, Conceptualization, Data curation, Formal analysis, Supervision, Funding acquisition, Validation, Investigation, Visualization, Methodology, Writing – original draft, Project administration, Writing – review and editing; Stefano Masserini, Conceptualization, Data curation, Software, Formal analysis, Validation, Visualization, Methodology, Writing – original draft, Project administration, Writing – review and editing; Laura Moreno Velasquez, Verjinia D Metodieva, Validation, Investigation, Writing – review and editing; Gaspar Cano, Conceptualization, Software, Validation, Writing – review and editing; Andrea Sannio, Data curation, Validation, Methodology, Writing – review and editing; Marta Orlando, Data curation, Funding acquisition, Validation, Methodology, Writing – review and editing; Nikolaus Maier, Conceptualization, Validation, Investigation, Methodology, Writing – review and editing; Richard Kempter, Conceptualization, Resources, Supervision, Funding acquisition, Validation, Project administration, Writing – review and editing; Dietmar Schmitz, Conceptualization, Supervision, Funding acquisition, Validation, Investigation, Project administration, Writing – review and editing

### Author ORCIDs

Rosanna P Sammons (D) http://orcid.org/0000-0001-9167-9263
Stefano Masserini (D) https://orcid.org/0009-0002-6672-0660
Laura Moreno Velasquez (D) http://orcid.org/0000-0001-9735-0039
Verjinia D Metodieva (D) http://orcid.org/0000-0003-1942-5140
Gaspar Cano (D) http://orcid.org/0000-0003-2076-1547
Andrea Sannio (D) https://orcid.org/0009-0005-0259-2411
Marta Orlando (D) http://orcid.org/0000-0002-9017-0251
Nikolaus Maier (D) https://orcid.org/0000-0001-5203-0736

Richard Kempter https://orcid.org/0000-0002-5344-2983
Dietmar Schmitz https://orcid.org/0000-0003-2741-5241

### Ethics

Animal maintenance and experiments were in accordance with the respective guidelines of local authorities (Berlin state government, T0100/03) and followed the German animal welfare act and the European Council Directive 2010/63/EU on protection of animals used for experimental and other scientific purposes.

Reviewer #1 (Public review): https://doi.org/10.7554/eLife.98653.3.sa1
Author response https://doi.org/10.7554/eLife.98653.3.sa2

## Additional files

### Supplementary files

MDAR checklist

Supplementary file 1. Parameter adjustments for model variants.

### Data availability

The code used to implement the computational model is available at https://github.com/stefano-masse/thorny_athorny (copy archived at *Masserini, 2025*). This repository includes the intermediate and final data generated by the code and displayed in the modeling figures.

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
