## [Editor Report · eLife Assessment]

This study represents **valuable** findings on the asymmetric connectivity pattern of two different types of CA3 pyramidal cell types showing that while athorny cells receive strong inputs from all other cell types, thorny cells receive weaker inputs from athorny neurons. Computational modeling is used to evaluate the impact of this connectivity scheme on the sequential activation of different cell types during sharp wave ripples. The evidence combining experimental and computational modelling approaches **convincingly** supports the authors' claims regarding the network mechanisms underlying the temporal sequences of neuronal activity during sharp-waves.

---

## [Referee Report · Reviewer #1 (Public review)]

Summary:

The hippocampal CA3 region is generally considered to be the primary site of initiation of sharp wave ripples-highly synchronous population events involved in learning and memory-although the precise mechanism remains elusive. A recent study revealed that CA3 comprises two distinct pyramidal cell populations: thorny cells that receive mossy fiber input from the dentate gyrus, and athorny cells that do not. That study also showed that it is athorny cells in particular which play a key role in sharp wave initiation. In the present work, Sammons, Masserini and colleagues expand on this by examining the connectivity probabilities among and between thorny and athorny cells. Using whole-cell patch clamp recordings, they find an asymmetrical connectivity pattern, with athorny cells receiving the most synaptic connections from both athorny and thorny cells, and thorny cells receiving fewer.

The authors then use a spiking network model to show how this assymmetrical connectivity is consistent with a preferential role of athorny cells in sharp wave initiation. Essentially, thorny and athorny cells are put into a winner-takes-all scenario in which athorny cells always win initially. Thorny cells can only become active after athorny cells decrease their firing rate due to adaptation, leading to a delay between the activation of athorny and thorny cells. As far as I understand, the initial victory of athorny cells in the winner-takes-all is doubly determined: it is both due to their intrinsic properties (lower rheobase and steeper f-I curve), and due to the bias in connectivity towards them. It appears to me that either of these two mechanisms (i.e., different intrinsic properties and symmetrical self- and cross-connections, or the same intrinsic properties and asymmetrical connectivity) would suffice to explain the sequential activation of the two cell types. From a theoretician's perspective, this overdetermination is not very elegant, but biology often isn't...

Strengths:

The authors provide independent validation of some of the findings by Hunt et al. (2018) concerning the distinction between thorny and athorny pyramidal cells in CA3 and advance our understanding of their differential integration in CA3 microcircuits. The properties of excitatory connections among and between thorny and athorny cells described by the authors will be key in understanding CA3 functions including, but not limited to, sharp wave initiation.

As stated in the paper, the modeling results lend support to the idea that the increased excitatory connectivity towards athorny cells plays an important role in causing them to fire before thorny cells in sharp waves. More generally, the model adds to an expanding pool of models of sharp wave ripples which should prove useful in guiding and interpreting experimental research.

---

## [Author Response]

The following is the authors’ response to the original reviews

**Public Reviews:**

**Reviewer #1 (Public Review):**
Summary:Sammons, Masserini et al. examine the connectivity of different types of CA3 pyramidal cells ("thorny" and "athorny"), and how their connectivity putatively contributes to their relative timing in sharp-wave-like activity. First, using patch-clamp recordings, they characterize the degree of connectivity within and between athorny and thorny cells. Based upon these experimental results, they compute a synaptic product matrix, and use this to inform a computational model of CA3 activity. This model finds that this differential connectivity between these populations, augmented by two different types of inhibitory neurons, can account for the relative timing of activity observed in sharp waves in vivo.

We thank the reviewer for reading our manuscript, as well as for their nice summary and constructive comments

Strengths:The patch-clamp experiments are exceptionally thorough and well done. These are very challenging experiments and the authors should be commended for their in-depth characterization of CA3 connectivity.

Thank you for the recognition of our efforts.

Weaknesses:(1) The computational elements of this study feel underdeveloped. Whereas the authors do a thorough job experimentally characterizing connections between excitatory neurons, the inhibitory neurons used in the model seem to be effectivity "fit neurons" and appear to have been tuned to produce the emergent properties of CA3 sharp wave-like activity. Although I appreciate the goal was to implicate CA3 connectivity contributions to activity timing, a stronger relationship seems like it could be examined. For example, did the authors try to "break" their model? It would be informative if they attempted different synaptic product matrices (say, the juxtaposition of their experimental product matrix) and see whether experimentally-derived sequential activity could not be elicited. It seems as though this spirit of analysis was examined in Figure 4C, but only insofar as individual connectivity parameters were changed in isolation.

Including the two interneuron types (B and C) in the model is, on the one hand, necessary to align our modeling framework to the state-of-the-art model by Evangelista et al. (2020), which assumes that these populations act as switchers between an SPW and a non-SPW state, and on the other hand, less straightforward because the connectivity involving these interneurons is largely unknown.

For B cells, the primary criterion to set their connections to and from excitatory cells was to balance the effect of the strong recurrent excitation and to achieve a mid-range firing rate for each population during sharp wave events. Our new simulations (Figure 5B) show that the initial suppression of population T (resulting in the long delay) indeed depends in equal proportions on the outlined excitatory connections and on how strongly each excitatory population is targeted by the B interneurons. However, these simulations demonstrate that there is a broad, clearly distinct, region of the parameter space that supports a long delay between the peaks, rather than a marginal set of finetuned parameters. In addition, the simulations show that B interneurons optimally contribute to the suppression of T when they primarily target T (Fig. 5B, panels 3,7,11,12,13) rather than A (panels 4,8,9,10,11). On the contrary, as reported in the parameter table, and now also displayed graphically in the new Figure 4A (included above, with arrow sizes proportional to the synaptic product between the parameters determining the total strength of each connection), we assume B to target A less weakly than T (to make up for the higher excitability of population A). Therefore, the long delay between the peaks in our model emerges in spite of the interneuron connectivity, rather than because of it, and it is an effect of the asymmetric connectivity between the two excitatory populations, in particular the extremely low connection from A to T.

(2) Additional explanations of how parameters for interneurons were incorporated in the model would be very helpful. As it stands, it is difficult to understand the degree to which the parameters of these neurons are biologically constrained versus used as fit parameters to produce different time windows of activity in types of CA3 pyramidal cells.

Response included in point (1).

**Reviewer #2 (Public Review):**
Sharp wave ripples are transient oscillations occurring in the hippocampus that are thought to play an important role in organising temporal sequences during the reactivation of neuronal activity. This study addresses the mechanism by which these temporal sequences are generated in the CA3 region focusing on two different subtypes of pyramidal neurons, thorny and athorny. Using high-quality electrophysiological recordings from up to 8 pyramidal neurons at a time the authors measure the connectivity rates between these pyramidal cell subtypes in a large dataset of 348 cells. This is a significant achievement and provides important data. The most striking finding is how similar connection characteristics are between cell types. There are no differences in synaptic strength or failure rates and some small differences in connectivity rates and short-term plasticity. Using model simulations, the authors explore the implications of the differences in connectivity rates for the temporal specificity of pyramidal cell firing within sharp-wave ripple events. The simulations show that the experimentally observed connectivity rates may contribute to the previously observed temporal sequence of pyramidal cell firing during sharp wave ripples.

Thank you very much for your careful review of our manuscript and the overall positive assessment.

The conclusions drawn from the simulations are not experimentally tested so remain theoretical. In the simple network model, the authors include basket cell and anti-SWR interneurons but the connectivity of these cell types is not measured experimentally and variations in interneuron parameters may also influence temporal specificity of firing.

As variations in some of these parameters can indeed influence the temporal specificity of firing, we have now performed additional simulations, the results of which are in the new Figures 5 and S5. Please also see response to Reviewer 1, point 1.

In addition, the influence of short-term plasticity measured in their experiments is not tested in the model.

We have now included short-term synaptic depression in all the excitatory-to-excitatory synapses and compensated for the weakened recurrent excitation by scaling some of the other parameters. The results of re-running our simulations in this alternative version of the model are reported in Figure S3 and are qualitatively analogous to those in Figure 4.

Interestingly, the experimental data reveal a large variability in many of the measured parameters. This may strongly influence the firing of pyramidal cells during SWRs but it is not represented within the model which uses the averaged data.

We have now incorporated variability in the following simulation parameters: the strength and latency of the four excitatory-to-excitatory connections as well as the reversal potential and leak conductance of both types of pyramidal cells, assuming variabilities similar to those observed experimentally (see Materials and Methods for details). Upon a slight re-balancing of some inhibitory connection strengths, in order to achieve comparable firing rates, we found that this version of the model also supports the generation of sharp waves with two pyramidal components (Figure S4B), and is, thus, fully analogous to our basic model. Varying the excitatory connectivities as in the original simulations (cf. Figure 4C and Figure S4C) reveals that increasing the athorny-toathorny or decreasing the athorny-to-thorny connectivity still increases the delay between the peaks, although for some connectivity values the peak of the athorny population appears more spread out in time.

**Reviewer #3 (Public Review):**
Summary:The hippocampal CA3 region is generally considered to be the primary site of initiation of sharp wave ripples-highly synchronous population events involved in learning and memory although the precise mechanism remains elusive. A recent study revealed that CA3 comprises two distinct pyramidal cell populations: thorny cells that receive mossy fiber input from the dentate gyrus, and athorny cells that do not. That study also showed that it is athorny cells in particular that play a key role in sharp wave initiation. In the present work, Sammons, Masserini, and colleagues expand on this by examining the connectivity probabilities among and between thorny and athorny cells. First, using whole-cell patch clamp recordings, they find an asymmetrical connectivity pattern, with athorny cells receiving the most synaptic connections from both athorny and thorny cells, and thorny cells receiving fewer. They then demonstrate in spiking neural network simulations how this asymmetrical connectivity may underlie the preferential role of athorny cells in sharp wave initiation.Strengths:The authors provide independent validation of some of the findings by Hunt et al. (2018) concerning the distinction between thorny and athorny pyramidal cells in CA3 and advance our understanding of their differential integration in CA3 microcircuits. The properties of excitatory connections among and between thorny and athorny cells described by the authors will be key in understanding CA3 functions including, but not limited to, sharp wave initiation.As stated in the paper, the modeling results lend support to the idea that the increased excitatory connectivity towards athorny cells plays a key role in causing them to fire before thorny cells in sharp waves. More generally, the model adds to an expanding pool of models of sharp wave ripples which should prove useful in guiding and interpreting experimental research.

Thank you very much for your careful review of our manuscript and this positive assessment.

Weaknesses:The mechanism by which athorny cells initiate sharp waves in the model is somewhat confusingly described. As far as I understood, random fluctuations in the activities of A and B neurons provide windows of opportunity for pyramidal cells to fire if they have additionally recovered from adaptive currents. Thorny and athorny pyramidal cells are then set in a winner-takes-all competition which is quickly won by the athorny cells. The main thesis of the paper seems to be that athorny cells win this competition because they receive more inputs both from themselves and from thorny cells, hence, the connectivity "underlies the sequential activation". However, it is also stated that athorny cells activate first due to their lower rheobase and steeper f-I curve, and it is also indicated in the methods that athorny (but not thorny) cells fire in bursts. It seems that it is primarily these features that make them fire first, something which apparently happens even when the A to A connectivity is set to 0albeit with a very small lag. Perhaps the authors could further clarify the differential role of single cell and network parameters in determining the sequential activation of athorny and thorny cells. Is the role of asymmetric excitatory connectivity only to enhance the initial intrinsic advantage of athorny cells? If so, could this advantage also be enhanced in other ways?

Thank you for the time invested in the review of our manuscript. We especially thank you for pointing out that the description of these dynamics was unclear: we have now improved it in the main text and we provide here an additional summary. As correctly highlighted by Reviewer 3, athorny neurons (A) are more excitable than thorny (T) ones due to single-neuron parameters: therefore, if there is a winner-takes-all competition, they are going to win it. Whether there is a competition in the first place, however, depends on the excitatory (and inhibitory) connections. In particular, we should distinguish two questions: does the activity of populations A and B (PV baskets), without adaptation (so at the beginning of the sharp wave) suppress T? And does the activity of populations T and B suppress A?

The four possible combinations can be appreciated, for example, in the new Figure 5A5. If A can suppress T, but T cannot suppress A (low A-to-T, high T-to-A, bottom right corner, like in the data), A “wins” and T fires later, after a long delay. If both A and T can suppress each other (both cross-connections are low, bottom left corner), we still get the same outcome: A wins because of its earlier and sharper onset (due to single-neuron parameters). If neither population can suppress the other (high cross-connections, top right corner), then there is no competition and the populations reach the peak approximately at the same time. Only in the case in which T can suppress A, but A cannot suppress T (low T-to-A, high A-to-T, top left corner, opposite to the data), then A “loses” the competition. However, since A neurons nevertheless display some early activity (again, due to the single neuron parameters), this scenario is not as clean as the reversed one: rather, A cells have an initial, small peak, then T neurons quickly take over and grow to their own peak, and then, depending on how strongly T neurons suppress A neurons, there may or may not be a second peak for the A neurons. This is the reason why, in the top left corner of Figure 5B, the statistics show either a long positive or long negative delay, depending on whether the first (small) or second (absent, for some parameters) peak of A is taken into account. In summary, the experimentally measured connectivity does not only enhance the initial intrinsic advantage of A cells, but sets up the competitive dynamics in the first place, which are crucial for the emergence of two distinct peaks, rather than a single peak involving both populations.

Although a clear effort has been made to constrain the model with biological data, too many degrees of freedom remain that allow the modeler to make arbitrary decisions. This is not a problem in itself, but perhaps the authors could explain more of their reasoning and expand upon the differences between their modeling choices and those of others. For example, what are the conceptual or practical advantages of using adaptation in pyramidal neurons as opposed to short-term synaptic plasticity as in the model by Hunt et al.?

It should be pointed out that the model by Hunt et al. features adaptation in pyramidal neurons as well, as the neuronal units employed are also adaptive-exponential integrate-and-fire. In an early stage of this project, we obtained from Hunt et al. the code for their model, and ascertained that adaptation is the main mechanism governing the alternations between the sharp-wave and the non-sharp-wave states, to the extent that fully removing short-term plasticity from their model does not have any significant impact on the network dynamics. Therefore, our choices are, in this regard, fully consistent with theirs. In order to confirm that synaptic depression does not significantly impact the dynamics also in our model, we now performed additional simulations (Figure S3), addressed in the main text (lines 149-151) and in the response to Reviewer 1, who expressed similar concerns.

Relatedly, what experimental observations could validate or falsify the proposed mechanisms?

As sharp wave generation in this model relies on disinhibitory dynamics (suppression of the anti-sharp-wave interneurons C), the model could be validated/falsified by proving/disproving that a class of interneurons with anti-sharp-wave features exists. In addition, the mechanism we proposed for the long delay between the peaks of the athorny and thorny activity requires at least some connectivity from athorny to basket and from basket to thorny neurons.

In the data by Hunt et al., thorny cells have a higher baseline (non-SPW) firing rate, and it is claimed that it is actually stochastic correlations in their firing that are amplified by athorny cells to initiate sharp waves. However, in the current model, the firing of both types of pyramidal cells outside of ripples appears to be essentially zero. Can the model handle more realistic firing rates as described by Hunt et al., or as produced by e.g., walking around an environment tiled with place cells, or would that trigger SPWs continuously?

When building this model, we aimed at having two clearly distinct states the network could alternate between, so we picked a rather polarized connectivity to and from the anti-sharp wave cells (C), resulting in polarized states. As a result, we obtain a low, although non-zero, activity of pyramidal neurons in non-SPW states (0.4 spikes/s for athorny and 0.2 spikes/s for thorny). These assumptions can be partially relaxed, for example in the original model by Evangelista et al. (2020), where the background firing rate of pyramidal cells is ~2 spikes/s. It should also be noted that, when walking in an environment tiled with place cells, the hippocampus is subject to additional extra-hippocampal inputs (e.g. from the medial septum, resulting in theta oscillations) and to neuromodulation, which can alter the network in various ways that we have not included in our model. However, our results are not in contradiction to transient SPW-like activity states initiated at a certain phase of the theta oscillation, when the inhibition is weakest.

**Recommendations for the authors:**

**Reviewer #1 (Recommendations For The Authors):**
(1) The manuscript reads like it was intended as a short-form manuscript for another journal. The introduction and discussion in particular are very brief and would benefit from being expanded and providing a bigger picture for the reader.

We had originally aimed to submit in the eLife “short report” format. However, also thanks to the suggestion of Reviewer 1, we realized that our text would be better supported by extended introduction and discussion sections, as well as additional figures.

(2) Graphs would benefit from including all datapoints, where appropriate.

All datapoints have now been added to boxplots in the main figures and supplement.

(3) The panels of Figure 4 are laid out strangely, it may be worthwhile to adjust.

We thank the reviewer for this suggestion. We have now adjusted the layout of Figure 4 and believe it is now easier to follow.

**Reviewer #2 (Recommendations For The Authors):**
Useful points to address include:(1) Explore within the model the effect of altering interneuron connectivity. Are there other factors that can influence temporal specificity within SWRs?

The effects of varying the connectivity to and from B interneurons (the ones which are SPWactive and therefore relevant for temporal specificity) have now been investigated in the new Figure 5B, in which such parameters were varied in pairs or combined with the two most relevant excitatoryto-excitatory connections.

(2) Implement the experimentally observed short-term plasticity in the model to determine how this influences temporal specificity.

All the findings in Figure 4 have now been replicated in the new Figure S3, in which excitatory-to-excitatory synapses feature synaptic depression.

(3) Consider if it is possible to incorporate observed experimental variability in the model and explore the implications.

All the findings in Figure 4 have now been replicated in the new Figure S4, in which heterogeneity has been introduced in multiple neuronal and synaptic parameters of thorny and athorny neurons.

(4) Include the co-connectivity rates in the data. Ie how many of the recorded neurons are reciprocally connected? Does this change the model simulations?

We have now added the rates of reciprocal connections that we observed into the main text (lines 86-88). We found 2 pairs of reciprocally connected athorny neurons and 2 pairs of reciprocally connected thorny neurons. These rates of reciprocity were not statistically significant. We did not observe reciprocal connections in other paired neuron combinations (i.e. athorny-thorny or vice-versa). Coconnectivity does not have any effect on the model simulations, as the model includes thousands of neurons grouped in populations without specific sub-structures. It might, however, be more relevant if the excitatory populations were further subdivided in assemblies.

**Reviewer #3 (Recommendations For The Authors):**
(1) Specify which part of CA3 you are recording from.

We have added this information into our results section - we recorded from 20 cells in CA3a, 274 cells in CA3b and 54 cells in CA3c. This information can now be found in the text on lines 68-69.

(2) Comment on why you might observe a larger fraction of athorny cells than Hunt et al.

Hunt et al. cite a broad range for the fraction of athorny cells in their discussion (10-20%). It is unclear where these estimates originate from. In their study, Hunt et al. use the bursting and nonbursting phenotypes as proxies for athorny and thorny cells respectively, and report here numbers of 32 and 70 equating to 31% athorny and 69% thorny. This fraction of athorny cells is more or less in line with our own findings, albeit slightly lower (34% and 66%). However, we believe this difference falls within the range of experimental variability. One caveat is that our electrophysiological recordings likely represent a biased sample of cells. In particular, with multipatch recordings, placement of later electrodes is often restricted to the borders of the pyramidal layer so as not to disturb already patched cells. Thus, our recorded cells do not represent a fully random sample of CA3 pyramidal cells. We believe that, only once a reliable genetic marker for athorny cells has been established can the size of this cell population be properly estimated. Furthermore, the ratio of thorny and athorny cells varies along the proximal distal axis of the CA3 so differences in ratios seen between our study and Hunt et al. may arise from sampling differences along this axis.

(3) In Figure 3, Aiii (the cell fractions) could also be represented as a vector of two squares stacked one on top of the other, then you could add multiplication signs between Ai, Aii and Aiii, and an equal sign between Aiii and Aiv.

Thank you! We have implemented this very nice suggestion.

(4) In Figure 4A, it would be helpful to display the strength of the connections similar to how it is done in Figure 3B.

We thank the reviewer for this suggestion. We have now updated Fig 4A to include connection strengths.